# PAC-BAYES INFORMATION BOTTLENECK

**Zifeng Wang** *
UIUC

**Shao-Lun Huang**
Tsinghua University

**Ercan E. Kuruoglu**
Tsinghua University

**Jimeng Sun**
UIUC

**Xi Chen**
Tencent

**Yefeng Zheng**
Tencent

## ABSTRACT

Understanding the source of the superior generalization ability of NNs remains one of the most important problems in ML research. There have been a series of theoretical works trying to derive non-vacuous bounds for NNs. Recently, the compression of information stored in weights (IIW) is proved to play a key role in NNs generalization based on the PAC-Bayes theorem. However, no solution of IIW has ever been provided, which builds a barrier for further investigation of the IIW's property and its potential in practical deep learning. In this paper, we propose an algorithm for the efficient approximation of IIW. Then, we build an IIW-based information bottleneck on the trade-off between accuracy and information complexity of NNs, namely PIB. From PIB, we can empirically identify the fitting to compressing phase transition during NNs' training and the concrete connection between the IIW compression and the generalization. Besides, we verify that IIW is able to explain NNs in broad cases, e.g., varying batch sizes, over-parameterization, and noisy labels. Moreover, we propose an MCMC-based algorithm to sample from the optimal weight posterior characterized by PIB, which fulfills the potential of IIW in enhancing NNs in practice.

## 1 INTRODUCTION

Understanding the behavior of neural networks (NNs) learned through stochastic gradient descent (SGD) is a prerequisite to revealing the source of NN's generalization in practice. Information bottleneck (IB) (Tishby et al., 2000) was a promising candidate to reveal the principle of NNs through the lens of information stored in encoded representations of inputs. Drawn from the conception of representation *minimality* and *sufficiency* in information theory, IB describes the objective of NNs as a trade-off, where NNs are abstracted by a Markov chain $Y \leftrightarrow X \leftrightarrow T$, as

$$\max_T I(T;Y) - \beta I(T;X), \tag{1}$$

where $I(T;X)$ and $I(T;Y)$ are the mutual information of representation $T$ towards inputs $X$ and labels $Y$, respectively. IB theory claimed (Tishby & Zaslavsky, 2015) and then empirically corroborated (Shwartz-Ziv & Tishby, 2017) that NNs trained by plain cross entropy loss and SGD all confront an initial fitting phase and a subsequent compression phase. It also implied that the representation compression is a causal effect to the good generalization capability of NNs. IB theorem points out the importance of representation compression and ignites a series of follow-ups to propose new learning algorithms on IB to explicitly take compression into account (Burgess et al., 2018; Achille & Soatto, 2018b; Dai et al., 2018; Li & Eisner, 2019; Achille & Soatto, 2018a; Kolchinsky et al., 2019; Wu et al., 2020; Pan et al., 2020; Goyal et al., 2019; Wang et al., 2019; 2020a).

However, recent critics challenged the universality of the above claims. At first, Saxe et al. (2019) argued that the representation compression phase only appears when *double-sided saturating nonlinearities* like `tanh` and `sigmoid` are deployed. The boundary between the two phases fades away with other nonlinearities, e.g., ReLU. Second, the claimed causality between compression and generalization was also questioned (Saxe et al., 2019; Goldfeld et al., 2019), i.e., sometime networks that do not compress still generalize well, and vice versa. To alleviate this issue, Goldfeld et al. (2019)

---

*Correspondence at zifengw2@illinois.edu

proposed that the clustering of hidden representations concurrently occurs with good generalization ability. However, this new proposal still lacks solid theoretical guarantee. Third, mutual information becomes trivial in deterministic cases (Shwartz-Ziv & Alemi, 2020). Other problems encountered in deterministic cases were pointed out by Kolchinsky et al. (2018). Although several techniques (Shwartz-Ziv & Tishby, 2017; Goldfeld et al., 2019), e.g., binning and adding noise, are adopted to make stochastic approximation for the information term, they might either violate the principle of IB or be contradictory to the high performance of NNs. Motivated by these developments, we focus on the following questions:

- Does a universal two-phase training behavior of NNs exist in practice? If this claim is invalid with the previous information measure $I(T; X)$, can we achieve this two-phase property through another information-theoretic perspective?
- As $I(T; X)$ was unable to fully explain NN's generalization, can we find another measure with theoretical generalization guarantee? Also, how do we leverage it to amend the IB theory for deep neural network?
- How do we utilize our new IB for efficient training and inference of NNs in practice?

In this work, we propose to handle the above questions through the lens of information stored in weights (IIW), i.e., $I(\mathbf{w}; S)$ where $S = \{X_i, Y_i\}_{i=1}^n$ is a finite-sample dataset. Our main contributions are four-fold: (1) we propose a new information bottleneck under the umbrella of PAC-Bayes generalization guarantee, namely **P**AC-Bayes **I**nformation **B**ottleneck (PIB); (2) we derive an approximation of the intractable IIW; (3) we design a Bayesian inference algorithm grounded on stochastic gradient Langevin dynamics (SGLD) for sampling from the optimal weight posterior specified by PIB; and (4) we demonstrate that our new information measure covers the wide ground of NN's behavior. Crediting to the plug-and-play modularity of SGD/SGLD, we can adapt any existing NN to a PAC-Bayes IB augmented NN seamlessly. Demo code is at https://github.com/RyanWangZf/PAC-Bayes-IB.

## 2 A New Bottleneck with PAC-Bayes Guarantee

In this section, we present the preliminaries of PAC-Bayes theory and then introduce our new information bottleneck. A loss function $\ell(f^{\mathbf{w}}(X), Y)$ is a measure of the degree of prediction accuracy $f^{\mathbf{w}}(X)$ compared with the ground-truth label $Y$. Given the ground-truth joint distribution $p(X, Y)$, the expected true risk (out-of-sample risk) is taken on expectation as

$$L(\mathbf{w}) \triangleq \mathbb{E}_{p(\mathbf{w}|S)}\mathbb{E}_{p(X,Y)}[\ell(f^{\mathbf{w}}(X), Y)]. \tag{2}$$

Note that we take an additional expectation over $p(\mathbf{w}|S)$ because we are evaluating risk of the learned posterior instead of a specific value of parameter $\mathbf{w}$. In addition, we call $p(\mathbf{w}|S)$ posterior here for convenience while it is not the Bayesian posterior that is computed through Bayes theorem $p(\mathbf{w}|S) = \frac{p(\mathbf{w})p(S|\mathbf{w})}{p(S)}$. The PAC-Bayes bounds hold even if prior $p(\mathbf{w})$ is incorrect and posterior $p(\mathbf{w}|S)$ is arbitrarily chosen.

In practice, we only own finite samples $S$. This gives rise to the empirical risk as

$$L_S(\mathbf{w}) = \mathbb{E}_{p(\mathbf{w}|S)}\left[\frac{1}{n}\sum_{i=1}^n \ell(f^{\mathbf{w}}(X_i), Y_i)\right]. \tag{3}$$

With the above Eqs. (2) and (3) at hand, the generalization gap of the learned posterior $p(\mathbf{w}|S)$ in out-of-sample test is $\Delta L(\mathbf{w}) \triangleq L(\mathbf{w}) - L_S(\mathbf{w})$. Xu & Raginsky (2017) proposed a PAC-Bayes bound based on the information contained in weights $I(\mathbf{w}; S)$ that

$$\mathbb{E}_{p(S)}[L(\mathbf{w}) - L_S(\mathbf{w})] \leq \sqrt{\frac{2\sigma^2}{n}I(\mathbf{w}; S)}, \tag{4}$$

when $\ell(f^{\mathbf{w}}(X), Y)$ is $\sigma$-sub-Gaussian.[1] A series of follow-ups tightened this bound and verified it is an effective measure of generalization capability of learning algorithms (Mou et al., 2018; Negrea

---

[1]A easy way to fulfill this condition is to clip the lost function to $\ell \in [0, a]$ hence it satisfies $\frac{a}{2}$-sub-Gaussian (Philippe, 2015; Xu & Raginsky, 2017).

et al., 2019; Pensia et al., 2018; Zhang et al., 2018). Therefore, it is natural to build a new information bottleneck grounded on this PAC-Bayes generalization measure, namely the PAC-Bayes information bottleneck (PIB), as

$$\min_{p(\mathbf{w}|S)} \mathcal{L}_{\text{PIB}} = L_S(\mathbf{w}) + \beta I(\mathbf{w}; S). \tag{5}$$

For classification term, the loss term $L_S(\mathbf{w})$ becomes the cross-entropy between the prediction $p(Y|X, \mathbf{w})$ and the label $p(Y|X)$, hence PIB in Eq. (5) is equivalent to

$$\max_{p(\mathbf{w}|S)} I(\mathbf{w}; Y|X, S) - \beta I(\mathbf{w}; S), \tag{6}$$

which demonstrates a trade-off between maximizing the *sufficiency* (the information of label $Y$ contained in $\mathbf{w}$) and minimizing the *minimality* of learned *parameters* $\mathbf{w}$ (the information of dataset $S$ contained in $\mathbf{w}$). Unlike previous IB based on *representations*, our PIB is built on *weights* that are not directly influenced by inputs and selected activation functions. Likewise, the trade-off described by PIB objective is more reasonable since its compression term is explicitly correlated to generalization of NNs.

## 3 ESTIMATING INFORMATION STORED IN WEIGHTS

In this section, we present a new notion of IIW $I(\mathbf{w}; S)$ built on the Fisher information matrix that relates to the flatness of the Riemannian manifold of loss landscape. Unlike Hessian eigenvalues of loss functions used for identifying flat local minima and generalization but can be made arbitrarily large (Dinh et al., 2017), this notion is invariant to re-parameterization of NNs (Liang et al., 2019). Also, our measure is invariant to the choice of activation functions because it is not directly influenced by input $X$ like $I(T; X)$. We leverage it to monitor the information trajectory of NNs trained by SGD and cross entropy loss and verify it is capable of reproducing the two-phase transition for varying activations (e.g., ReLU, linear, tanh, and sigmoid) in §5.1.

### 3.1 CLOSED-FORM SOLUTION WITH GAUSSIAN ASSUMPTION

By deriving a new information bottleneck PIB, we can look into how IIW $I(\mathbf{w}; S)$ and $L_S(\mathbf{w})$ evolve during the learning process of NNs optimized by SGD. Now the key challenge ahead is how to estimate the IIW $I(\mathbf{w}; S)$, as

$$I(\mathbf{w}; S) = \mathbb{E}_{p(S)}[\text{KL}(p(\mathbf{w}|S) \| p(\mathbf{w}))] \tag{7}$$

is the expectation of Kullback-Leibler (KL) divergence between $p(\mathbf{w}|S)$ and $p(\mathbf{w})$ over the distribution of dataset $p(S)$. And, $p(\mathbf{w})$ is the marginal distribution of $p(\mathbf{w}|S)$ as $p(\mathbf{w}) \triangleq \mathbb{E}_{p(S)}[p(\mathbf{w}|S)]$. When we assume both $p(\mathbf{w}) = \mathcal{N}(\mathbf{w}|\boldsymbol{\theta}_0, \Sigma_0)$ and $p(\mathbf{w}|S) = \mathcal{N}(\mathbf{w}|\boldsymbol{\theta}_S, \Sigma_S)$ are Gaussian distributions, the KL divergence term in Eq. (7) has closed-form solution as

$$\text{KL}(p(\mathbf{w}|S) \| p(\mathbf{w})) = \frac{1}{2} \left[ \log \frac{\det \Sigma_S}{\det \Sigma_0} - D + (\boldsymbol{\theta}_S - \boldsymbol{\theta}_0)^\top \Sigma_0^{-1} (\boldsymbol{\theta}_S - \boldsymbol{\theta}_0) + \text{tr} \left( \Sigma_0^{-1} \Sigma_S \right) \right]. \tag{8}$$

Here $\det \mathbf{A}$ and $\text{tr}(\mathbf{A})$ are the determinant and trace of matrix $\mathbf{A}$, respectively; $D$ is the dimension of parameter $\mathbf{w}$ and is a constant for a specific NN architecture; $\boldsymbol{\theta}_S$ are the yielded weights after SGD converges on the dataset $S$. If the covariances of prior and posterior are proportional,[2] the logarithmic and trace terms in Eq. (8) both become constant. Therefore, the mutual information term is proportional to the quadratic term as

$$I(\mathbf{w}; S) \propto \mathbb{E}_{p(S)} \left[ (\boldsymbol{\theta}_S - \boldsymbol{\theta}_0)^\top \Sigma_0^{-1} (\boldsymbol{\theta}_S - \boldsymbol{\theta}_0) \right] = \mathbb{E}_{p(S)} \left[ \boldsymbol{\theta}_S^\top \Sigma_0^{-1} \boldsymbol{\theta}_S \right] - \boldsymbol{\theta}_0^\top \Sigma_0^{-1} \boldsymbol{\theta}_0. \tag{9}$$

In the next section, we will see how to set prior covariance $\Sigma_0$.

---

[2]Assuming the same covariance for the Gaussian randomization of posterior and prior is a common practice of building PAC-Bayes bound for simplification, see (Dziugaite & Roy, 2018; Rivasplata et al., 2018).

## 3.2 BOOTSTRAP COVARIANCE OF ORACLE PRIOR

Since the computation of exact oracle prior $\Sigma_0$ needs the knowledge of $p(S)$ [3], we propose to approximate it by *bootstrapping* from $S$ as

$$\Sigma_0 = \mathbb{E}_{p(S)}\left[(\boldsymbol{\theta}_S - \boldsymbol{\theta}_0)(\boldsymbol{\theta}_S - \boldsymbol{\theta}_0)^\top\right] \simeq \frac{1}{K}\sum_k (\boldsymbol{\theta}_{S_k} - \boldsymbol{\theta}_S)(\boldsymbol{\theta}_{S_k} - \boldsymbol{\theta}_S)^\top, \tag{10}$$

where $S_k$ is a bootstrap sample obtained by re-sampling from the finite data $S$, and $S_k \sim p(S)$ is still a valid sample following $p(S)$. Now we are closer to the solution but the above term is still troublesome to calculate. For getting $\{\theta_{S_k}\}_{k=1}^K$, we need to optimize on a series of ($K$ times) bootstrapping datasets $\{S_k\}_{k=1}^K$ via SGD until it converges, which is prohibitive in deep learning practices. Therefore, we propose to approximate the difference $\boldsymbol{\theta}_S - \boldsymbol{\theta}_0$ by *influence functions* drawn from robust statistics literature (Cook & Weisberg, 1982; Koh & Liang, 2017; Wang et al., 2020c;b).

**Lemma 1** (Influence Function (Cook & Weisberg, 1982)). *Given a dataset $S = \{(X_i, Y_i)\}_{i=1}^n$ and the parameter $\hat{\boldsymbol{\theta}}_S \triangleq \operatorname{argmin}_{\boldsymbol{\theta}} L_S(\boldsymbol{\theta}) = \operatorname{argmin}_{\boldsymbol{\theta}} \frac{1}{n}\sum_{i=1}^n \ell_i(\boldsymbol{\theta})$[4] that optimizes the empirical loss function, if we drop sample $(X_j, Y_j)$ in $S$ to get a jackknife sample $S_{\backslash j}$ and retrain our model, the new parameters are $\hat{\boldsymbol{\theta}}_{S_{\backslash j}} = \operatorname{argmin}_{\boldsymbol{\theta}} L_{S_{\backslash j}}(\boldsymbol{\theta}) = \operatorname{argmin}_{\boldsymbol{\theta}} \frac{1}{n}\sum_{i=1}^n \ell_i(\boldsymbol{\theta}) - \frac{1}{n}\ell_j(\boldsymbol{\theta})$. The approximation of parameter difference $\hat{\boldsymbol{\theta}}_{S_{\backslash j}} - \hat{\boldsymbol{\theta}}_S$ is defined by influence function $\boldsymbol{\psi}$, as*

$$\hat{\boldsymbol{\theta}}_{S_{\backslash j}} - \hat{\boldsymbol{\theta}}_S \simeq -\frac{1}{n}\boldsymbol{\psi}_j, \text{ where } \boldsymbol{\psi}_j = -\mathbf{H}_{\hat{\boldsymbol{\theta}}_S}^{-1}\nabla_{\boldsymbol{\theta}}\ell_j(\hat{\boldsymbol{\theta}}_S) \in \mathbb{R}^D, \tag{11}$$

*and $\mathbf{H}_{\hat{\boldsymbol{\theta}}_S} \triangleq \frac{1}{n}\sum_{i=1}^n \nabla_{\boldsymbol{\theta}}^2 \ell_i(\hat{\boldsymbol{\theta}}_S) \in \mathbb{R}^{D\times D}$ is Hessian matrix.*

The application of influence functions can be further extended to the case when the loss function is perturbed by a vector $\boldsymbol{\xi} = (\xi_1, \xi_2, \ldots, \xi_n)^\top \in \mathbb{R}^n$ as $\hat{\boldsymbol{\theta}}_{S,\boldsymbol{\xi}} = \operatorname{argmin}_{\boldsymbol{\theta}} \frac{1}{n}\sum_{i=1}^n \xi_i\ell_i(\boldsymbol{\theta})$. In this scenario, the parameter difference can be approximated by

$$\hat{\boldsymbol{\theta}}_{S,\boldsymbol{\xi}} - \hat{\boldsymbol{\theta}}_S \simeq \frac{1}{n}\sum_{i=1}^n (\xi_i - 1)\boldsymbol{\psi}_i = \frac{1}{n}\Psi^\top(\boldsymbol{\xi} - \mathbf{1}), \tag{12}$$

where $\Psi = (\boldsymbol{\psi}_1, \boldsymbol{\psi}_2, \ldots, \boldsymbol{\psi}_n)^\top \in \mathbb{R}^{n\times D}$ is a combination of all influence functions $\boldsymbol{\psi}$; $\mathbf{1} = (1, 1, \ldots, 1)^\top$ is an $n$-dimensional all-one vector. Lemma 1 gives rise to the following lemma on approximation of oracle prior covariance in Eq. (10):

**Lemma 2** (Approximation of Oracle Prior Covariance). *Given the definition of influence functions (Lemma 1) and Poisson bootstrapping (Lemma A.2), the covariance matrix of the oracle prior can be approximated by*

$$\Sigma_0 = \mathbb{E}_{p(S)}\left[(\boldsymbol{\theta}_S - \boldsymbol{\theta}_0)(\boldsymbol{\theta}_S - \boldsymbol{\theta}_0)^\top\right] \simeq \frac{1}{K}\sum_{k=1}^K \left(\hat{\boldsymbol{\theta}}_{\boldsymbol{\xi}^k} - \hat{\boldsymbol{\theta}}\right)\left(\hat{\boldsymbol{\theta}}_{\boldsymbol{\xi}^k} - \hat{\boldsymbol{\theta}}\right)^\top \simeq \frac{1}{n}\mathbf{H}_{\hat{\boldsymbol{\theta}}}^{-1}\mathbf{F}_{\hat{\boldsymbol{\theta}}}\mathbf{H}_{\hat{\boldsymbol{\theta}}}^{-1} \simeq \frac{1}{n}\mathbf{F}_{\hat{\boldsymbol{\theta}}}^{-1}, \tag{13}$$

*where $\mathbf{F}_{\hat{\boldsymbol{\theta}}}$ is Fisher information matrix (FIM); we omit the subscript $S$ of $\hat{\boldsymbol{\theta}}_S$ and $\hat{\boldsymbol{\theta}}_{S,\boldsymbol{\xi}}$ for notation conciseness, and $\boldsymbol{\xi}^k$ is the bootstrap resampling weight in the $k$-th experiment.*

Please refer to Appendix A for the proof of this lemma.

## 3.3 EFFICIENT INFORMATION ESTIMATION ALGORITHM

After the approximation of oracle prior covariance, we are now able to rewrite the IIW term $I(\mathbf{w}; S)$ in Eq. (9) to

$$I(\mathbf{w}; S) \propto n\mathbb{E}_{p(S)}\left[(\boldsymbol{\theta}_S - \boldsymbol{\theta}_0)^\top \mathbf{F}_{\hat{\boldsymbol{\theta}}}(\boldsymbol{\theta}_S - \boldsymbol{\theta}_0)\right] \simeq n(\bar{\boldsymbol{\theta}}_S - \boldsymbol{\theta}_0)^\top \mathbf{F}_{\hat{\boldsymbol{\theta}}}(\bar{\boldsymbol{\theta}}_S - \boldsymbol{\theta}_0) = \widetilde{I}(\mathbf{w}; S). \tag{14}$$

---

[3] $\Sigma_0$ is called oracle prior because it minimizes the term $I(\mathbf{w}; S) = \mathbb{E}_{p(S)}[\mathrm{KL}(p(w|S) \| p(W))]$ when $p(w) = \mathbb{E}_{p(S)}[p(w|S)]$ (Dziugaite et al., 2021).

[4] Note $L_S(\boldsymbol{\theta})$ is not the expected empirical risk $L_S(\mathbf{w})$ in Eq. (3); instead, it is the deterministic empirical risk that only relates to the mean parameter $\boldsymbol{\theta}$. We also denote $\ell(f^{\boldsymbol{\theta}}(X_i), Y_i)$ by $\ell_i(\boldsymbol{\theta})$ for the notation conciseness.

---

**Algorithm 1:** Efficient approximate information estimation of $I(\mathbf{w}; S)$

---

**Data:** Total number of samples $n$, batch size $B$, number of mini-batches in one epoch $T_0$, number of information estimation iterations $T_1$, learning rate $\eta$, moving average hyperparameters $\rho$ and $K$, a sequence of gradients set $\nabla\mathcal{L} = \emptyset$

**Result:** Calculated approximate information $\widetilde{I}(\mathbf{w}; S)$

1 Pretrain the model by vanilla SGD to obtain the prior mean $\boldsymbol{\theta}_0$ ;
2 **for** *t=1:$T_0$* **do**
3 $\quad$ $\nabla L_t \leftarrow \nabla_{\boldsymbol{\theta}} \frac{1}{B}\sum_b \ell_b(\hat{\boldsymbol{\theta}}_{t-1}), \hat{\boldsymbol{\theta}}_t \leftarrow \hat{\boldsymbol{\theta}}_{t-1} - \eta\nabla L_t$ ; $\qquad\qquad$ /* Vanilla SGD */
4 $\quad$ $\nabla\mathcal{L} \leftarrow \nabla\mathcal{L} \bigcup\{\nabla L_t\}$ ; $\qquad\qquad\qquad\qquad$ /* Store gradients */
5 $\quad$ $\bar{\boldsymbol{\theta}}_t \leftarrow \sqrt{\rho\bar{\boldsymbol{\theta}}_{t-1}^2 + \frac{1-\rho}{K}\sum_{k=0}^{K-1}\hat{\boldsymbol{\theta}}_{t-k}^2}$ ; $\qquad\qquad$ /* Moving average */
6 **end**
7 $\Delta\boldsymbol{\theta} \leftarrow \bar{\boldsymbol{\theta}}_{T_0} - \boldsymbol{\theta}_0, \ \Delta\mathbf{F}_0 \leftarrow 0$ ;
8 **for** *t=1:$T_1$* **do**
9 $\quad$ $\Delta\mathbf{F}_t \leftarrow \Delta\mathbf{F}_{t-1} + (\Delta\boldsymbol{\theta}^\top\nabla L_t)^2$ ; $\qquad$ /* Storage-friendly computation */
10 **end**
11 $\widetilde{I}(\mathbf{w}; S) \leftarrow \frac{n}{T_1}\Delta\mathbf{F}_{T_1}$;

---

**Algorithm 2:** Optimal Gibbs posterior inference by SGLD.

---

**Data:** Total number of samples $n$, batch size $B$, learning rate $\eta$, temperature $\beta$

**Result:** A sequence of weights $\{\mathbf{w}_t\}_{t\geq\hat{k}}$ following $p(\mathbf{w}|S^*)$

1 **repeat**
$\quad$ /* Mini-batch gradient of energy function $\qquad\qquad\qquad\qquad$ */
2 $\quad$ $\nabla\widetilde{U}_{S^*}(\mathbf{w}_{t-1}) \leftarrow \nabla\left(-\frac{B}{n}\sum_b \log p(Y_b|X_b, \mathbf{w}_{t-1}) - \beta_{t-1}\log p(\mathbf{w}_{t-1})\right)$ ;
$\quad$ /* SGLD by gradient plus isotropic Gaussian noise $\qquad\qquad$ */
3 $\quad$ $\varepsilon_t \leftarrow \mathcal{N}(\varepsilon|\mathbf{0}, \mathbf{I}_D), \mathbf{w}_t \leftarrow \mathbf{w}_{t-1} - \eta_{t-1}\nabla\widetilde{U}_{S^*}(\mathbf{w}_{t-1}) + \sqrt{2\eta_{t-1}\beta_{t-1}}\varepsilon_t$ ;
$\quad$ /* Learning rate & temperature decay $\qquad\qquad\qquad\qquad\qquad$ */
4 $\quad$ $\eta_t \leftarrow \phi_\eta(\eta_{t-1}), \beta_t \leftarrow \phi_\beta(\beta_{t-1}), t \leftarrow t+1$ ;
5 **until** *the weight sequence* $\{\mathbf{w}_t\}_{t\geq\hat{k}}$ *becomes stable*;

---

We define the approximate information by $\widetilde{I}(\mathbf{w}; S)$ where we approximate the expectation $\mathbb{E}_{p(S)}[\boldsymbol{\theta}_S^\top F_{\hat{\boldsymbol{\theta}}}\boldsymbol{\theta}_S]$ by $\bar{\boldsymbol{\theta}}_S = \sqrt{\frac{1}{K}\sum_{k=1}^K \hat{\boldsymbol{\theta}}_k^2} = \left(\sqrt{\frac{1}{K}\sum_{k=1}^K\hat{\theta}_{1,k}^2}, \ldots, \sqrt{\frac{1}{K}\sum_{k=1}^K\hat{\theta}_{D,k}^2}\right)^\top$.[5] In Eq. (14), the information consists of two major components: $\Delta\boldsymbol{\theta} = \bar{\boldsymbol{\theta}}_S - \boldsymbol{\theta}_0 \in \mathbb{R}^D$ and $\mathbf{F}_{\hat{\boldsymbol{\theta}}} \in \mathbb{R}^{D\times D}$, which can easily cause out-of-memory error due to the high-dimensional matrix product operations. We therefore hack into FIM to get

$$\widetilde{I}(\mathbf{w}; S) = n\Delta\boldsymbol{\theta}^\top\left[\frac{1}{T}\sum_{t=1}^T \nabla_{\boldsymbol{\theta}}\ell_t(\hat{\boldsymbol{\theta}})\nabla_{\boldsymbol{\theta}}\ell_t^\top(\hat{\boldsymbol{\theta}})\right]\Delta\boldsymbol{\theta} = \frac{n}{T}\sum_{t=1}^T\left[\Delta\boldsymbol{\theta}^\top\nabla_{\boldsymbol{\theta}}\ell_t(\hat{\boldsymbol{\theta}})\right]^2, \qquad (15)$$

such that the high dimensional matrix vector product reduces to vector inner product. We encapsulate the algorithm for estimating IIW during vanilla SGD by Algorithm 1.

## 4 BAYESIAN INFERENCE FOR THE OPTIMAL POSTERIOR

Recall that we designed a new bottleneck on the expected generalization gap drawn from PAC-Bayes theory in §2, and then derived an approximation of the IIW in §3. The two components of PAC-Bayes IB in Eq. (6) are hence tractable as a learning objective. We give the following lemma on utilizing it for inference.

---

[5]The quadratic mean is closer to the true value than arithmetic mean because of the quadratic term within the expectation function.

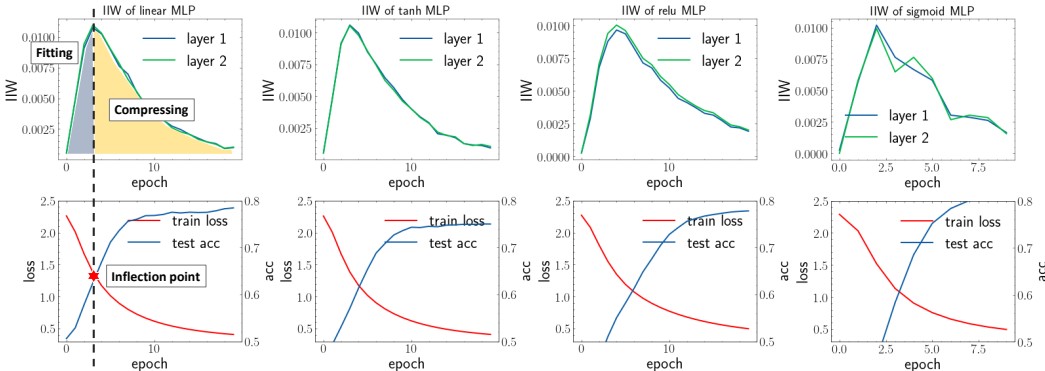

Figure 1: IIW (up), loss and accuracy (down) of different activation functions (`linear`, `tanh`, `ReLU`, and `sigmoid`) NNs. There is a clear boundary between the initial **fitting** and the **compression** phases identified by IIW. Meanwhile, the train loss encounters the inflection point that keeps decreasing with slower slope. Note that the learning rate is set small (1e-4) except for `sigmoid`-NN for the better display of two phases.

**Lemma 3** (Optimal Posterior for PAC-Bayes Information Bottleneck). *Given an observed dataset $S^*$, the optimal posterior $p(\mathbf{w}|S^*)$ of PAC-Bayes IB in Eq.* (5) *should satisfy the following form that*

$$p(\mathbf{w}|S^*) = \frac{1}{Z(S)} p(\mathbf{w}) \exp\left\{-\frac{1}{\beta}\hat{L}_{S^*}(\mathbf{w})\right\} = \frac{1}{Z(S)} \exp\left\{-\frac{1}{\beta}U_{S^*}(\mathbf{w})\right\}, \quad (16)$$

*where $U_{S^*}(\mathbf{w})$ is the energy function defined as $U_{S^*}(\mathbf{w}) = \hat{L}_{S^*}(\mathbf{w}) - \beta \log p(\mathbf{w})$, and $Z(S)$ is the normalizing constant.*

Please refer to Appendix B for the proof. The reason why we write the posterior in terms of an exponential form is that it is a typical *Gibbs distribution* (Kittel, 2004) (also called Boltzmann distribution) with *energy function $U_{S^*}(\mathbf{w})$* and *temperature $\beta$* (the same $\beta$ of PIB appears in Eq. (5)). Crediting to this formula, we can adopt Markov chain Monte Carlo (MCMC) for rather efficient Bayesian inference. Specifically, we propose to use stochastic gradient Langevin dynamics (SGLD) (Welling & Teh, 2011) that has been proved efficient and effective in large-scale posterior inference. SGLD can be realized by a simple adaption of SGD as

$$\mathbf{w}_{k+1} = \mathbf{w}_k - \eta_k \mathbf{g}_k + \sqrt{2\eta_k \beta}\varepsilon_k, \quad (17)$$

where $\eta_k$ is step size, $\varepsilon_k \sim \mathcal{N}(\varepsilon|\mathbf{0}, \mathbf{I}_D)$ is a standard Gaussian noise vector, and $\mathbf{g}_k$ is an unbiased estimate of energy function gradient $\nabla U(\mathbf{w}_k)$. SGLD can be viewed as a discrete Langevin diffusion described by stochastic differential equation (Raginsky et al., 2017; Borkar & Mitter, 1999): $d\mathbf{w}(t) = -\nabla U(\mathbf{w}(t))dt + \sqrt{2\beta}dB(t)$, where $\{B(t)\}_{t\geq 0}$ is the standard Brownian motion in $\mathbb{R}^D$. The Gibbs distribution $\pi(\mathbf{w}) \propto \exp(-\frac{1}{\beta}U(\mathbf{w}))$ is the unique invariant distribution of the Langevin diffusion. And, distribution of $\mathbf{w}_t$ converges rapidly to $\pi(\mathbf{w})$ when $t \to \infty$ with sufficiently small $\beta$ (Chiang et al., 1987). Similarly for SGLD in Eq. (17), under the conditions that $\sum_t^\infty \eta_t \to \infty$ and $\sum_t^\infty \eta_t^2 \to 0$, and an annealing temperature $\beta$, the sequence of $\{\mathbf{w}_k\}_{k\geq \hat{k}}$ converges to Gibbs distribution with sufficiently large $\hat{k}$.

As we assume the oracle prior $p(\mathbf{w}) = \mathcal{N}(\mathbf{w}|\boldsymbol{\theta}_0, \Sigma_0)$, $\log p(\mathbf{w})$ satisfies

$$-\log p(\mathbf{w}) \propto (\mathbf{w} - \boldsymbol{\theta}_0)^\top \Sigma_0^{-1}(\mathbf{w} - \boldsymbol{\theta}_0) + \log(\det \Sigma_0). \quad (18)$$

The inference of the optimal posterior is then summarized by Algorithm 2. $\phi_\eta(\cdot)$ and $\phi_\beta(\cdot)$ are learning rate decay and temperature annealing functions, e.g., cosine decay, respectively. Our SGLD based algorithm leverages the advantage of MCMC such that it is capable of sampling from the optimal posterior even for very complex NNs. Also, it does need to know the groundtruth distribution of $S$ while still theoretically allows global convergence avoiding local minima. It can be realized with a minimal adaptation of common auto-differentiation packages, e.g., PyTorch (Paszke et al., 2019), by injecting isotropic noise in the SGD updates. Please refer to Appendix C for the details of computation for PIB object.

## 5 EXPERIMENTS

In this section, we aim to verify the intepretability of the proposed notion of IIW by Eq. (15). We monitor the information trajectory when training NNs **with plain cross entropy loss and SGD** for the sake of activation functions (§5.1), architecture (§5.2), noise ratio (§5.3), and batch size (§5.4). We also substantiate the superiority of optimal Gibbs posterior inference based on the proposed Algorithm 2, where PIB instead of plain cross entropy is used as the objective function (§5.5). We conclude the empirical observations in §5.6 at last. Please refer to Appendix D for general experimental setups about the used datasets and NNs.

### 5.1 INFORMATION WITH DIFFERENT ACTIVATION FUNCTIONS

We train a 2-layer MLP (784-512-10) with plain cross-entropy loss by Adam on the MNIST dataset, meanwhile monitor the trajectory of the IIW $I(\mathbf{w}; S)$. Results are illustrated in Fig. 1 where different activation functions (linear, tanh, ReLU, and sigmoid) are testified. We identify that there is a clear boundary between fitting and compression phases for all of them. For example, for the linear activation function on the first column, the IIW $I(\mathbf{w}; S)$ surges within the first several iterations then drops slowly during the next iterations. Simultaneously, we could see that the training loss reduces sharply at the initial stage, then keeps decreasing during the information compression. At the last period of compression, we witness the information fluctuates near zero with the recurrence of memorization phenomenon (IIW starts to increase). This implies

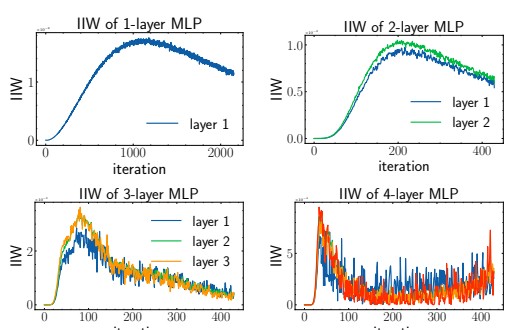

Figure 2: Information compression with varying number of layers (1, 2, 3, and 4) for ReLU MLPs. All follow the general trend of fitting to compression phase transition. And, deeper layers can accelerate both the fitting and compressing phases.

that further training is causing over-fitting. IIW shows great universality on representing the information compression of NNs.

### 5.2 INFORMATION WITH DEEPER AND WIDER ARCHITECTURE

Having identified the phase transition of the 2-layer MLP corresponding to IIW $I(\mathbf{w}; S)$, we test it under more settings: different architectures and different batch sizes. For the architecture setting, we design MLPs from 1 to 4 layers. Results are shown in Fig. 2. The first and the last figures show the information trajectory of the 1-layer/4-layer version of MLP-Large (784-10/784-512-100-80-10) where clear two-phase transitions happen in all these MLPs.

The 1-layer MLP is actually a softmax regression model. It can be identified that this model fits and compresses very slowly w.r.t. IIW compared with deeper NNs. This phenomenon demonstrates that deep models have overwhelmingly learning capacity than shallow models, because deep layers can not only boost the memorization of data but also urges the model to compress the redundant information to gain better generalization ability. Furthermore, when we add more layers, the fitting phase becomes shorter. Specifically, we observe the incidence of overfitting at the end of the 4-layer MLP training as IIW starts to increase.

We also examine how IIW explains the generalization w.r.t. the number of hidden units, a.k.a. the width of NNs, by Fig. 3. We train a 2-layer MLP without any regularization on MNIST. The left panel shows the training and test errors for this experiment. Notably, the difference of test and train acc can be seen an indicator of the **generalization gap** in Eq. (4). IIW should be aligned to this gap by definition. While 32 units are (nearly) enough to interpolate the training set, more hidden units still achieve better generalization performance, which illustrates the effect of overparameterization. In this scenario, weights $\ell_2$-norm keeps increasing with more units while IIW decays, similar to the test error. We identify that more hidden units do not render much increase of IIW, which is contrast

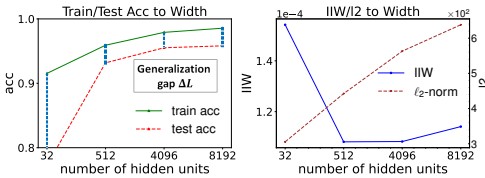 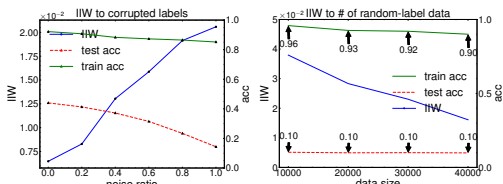

Figure 3: **Left**: Training and test accuracy w.r.t. # units; **Right**: Complexity measure (IIW and $\ell_2$-norm) w.r.t. # units. Blue dash line shows the gap between train/test acc (generalization gap). We find $\ell_2$-norm keeps increasing with more hidden units. Instead, IIW keeps pace with the generalization gap: the larger the gap, the larger the IIW.

Figure 4: **Left**: IIW, train, and test accuracy when noise ratio in labels changes. IIW rises when noise ratio grows. **Right**: IIW with varying size of random-label data. Test acc keeps constant while train acc decays. Hence, more data causes lower IIW because of the shrinking gap between the train and test accuracy.

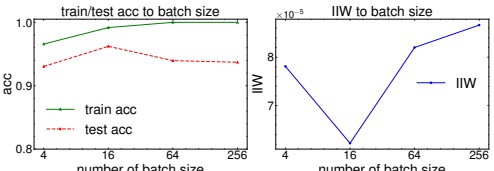 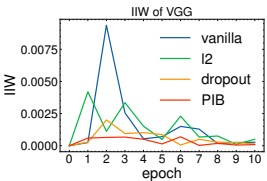

Figure 5: **Left**: Training and test accuracy w.r.t. # batch size; **Right**: IIW w.r.t. # batch size. We find IIW keeps pace with the generalization gap: the larger the gap, the larger the IIW. From IIW we can specify that 16 is the best which reaches the least generalization gap without the need of having the model tested.

Figure 6: The tracked IIW of the VGG net during the training by four ways: vanilla, $\ell_2$-norm regularization, dropout, and PIB training. We can identify that: first, all of them still follow a fitting-compressing paradigm specified by IIW; second, vanilla VGG reaches the largest IIW far above the others; third, PIB regularizes IIW directly thus yielding the smallest IIW.

to the intuition that wider NNs always have larger information complexity. More importantly, we find IIW is consistent to the generalization gap on each width.

## 5.3 RANDOM LABELS VS. TRUE LABELS

According to the PAC-Bayes theorem, the IIW is a promising measure to explain/predict the generalization capability of NNs. NNs are often over-parameterized thus can perfectly fit even the random labels, obviously without any generalization capability. For example, 2-layer MLP has $15,728,640$ parameters that are much larger than the sample number of CIFAR-10 (50,000). That causes the number of parameters an unreliable measure of NN complexity in overparameterization settings (Neyshabur et al., 2015). Alternatively, $\ell_2$-norm is often used as a complexity measure to be imposed on regularizing model training in practices.

We investigate the model trained with different levels of label corruption, as shown by the left panel of Fig. 4. We train a 2-layer MLP on CIFAR-10 and find that the increasing noise causes sharp test acc decay while train acc does not change much. Meanwhile, IIW keeps growing with the fall of test acc and expansion of generalization gap. This demonstrates IIW's potential in identifying the noise degree in datasets or the mismatch between the test and train data distributions.

We further build a random-label CIFAR-10, results are on the right of Fig. 4. Although the model can still nearly interpolate the train data, with the rise of random-label data, the model keeps 10% test acc but has less train acc, which renders larger generalization gap. This phenomenon is also captured by IIW.

## 5.4 INFORMATION IN WEIGHTS W.R.T. BATCH SIZE

We also consider how batch size influences IIW and generalization gap. Recent efforts on bounding $I(\mathbf{w}; S)$ of iterative algorithms (e.g., SGD and SGLD) (Mou et al., 2018; Pensia et al., 2018) imply that the variance of gradients is a crucial factor. For the ultimate case where batch size equals

Table 1: Test performance of the proposed PIB algorithm compared with two other common regularization techniques: $\ell_2$-norm and dropout, on VGG-net (Simonyan & Zisserman, 2014). The 95% confidence intervals are shown in parentheses. Best values are in bold.

| Test ACC (%) | CIFAR10 | CIFAR100 | STL10 | SVHN |
|---|---|---|---|---|
| vanilla SGD | 77.03(0.57) | 52.07(0.44) | 54.31(0.65) | 93.57(0.67) |
| SGD+$\ell_2$-norm | 77.13(0.53) | 50.84(0.71) | 55.30(0.68) | 93.60(0.68) |
| SGD+dropout | 78.95(0.60) | 52.34(0.66) | 56.35(0.78) | 93.61(0.76) |
| SGD+PIB | **80.19(0.42)** | **56.47(0.62)** | **58.83(0.75)** | **93.88(0.88)** |

full sample size, the gradient variance is zero, and the model is prone to over-fitting grounded on empirical observations. When batch size equals one, the variance becomes tremendously large.

We conjecture there is an optimal batch size that reaches the minimum generalization gap, in other word, the minimum IIW. This conjecture is raised on our empirical findings, displayed in Fig. 5 where we test IIW on model with varying batch size. Each model is updated with the same total number of iterations and the same learning rate. We identify that when batch size is 16, the model reaches the best test acc and the least generalization gap, which means this optimal batch size should fall into (4,16) or (16, 64). On the left, the model reaches the minimum IIW when batch size is 16.

## 5.5 BAYESIAN INFERENCE WITH VARYING ENERGY FUNCTIONS

To confirm the superiority of the proposed PIB in §4, we compare it with vanilla SGD and two widely used regularizations: $\ell_2$-norm and dropout. We train a large VGG network (Simonyan & Zisserman, 2014) on four open datasets: CIFAR10/100 (Krizhevsky et al., 2009), STL10 (Coates et al., 2011), and SVHN (Netzer et al., 2011), as shown in Table 1, where we find PIB consistently outperforms the baselines. We credit the improvement to the explicit consideration of information regularization during the training, which forces the model to *forget* the training dataset to regularize the generalization gap. This is verified by Fig. 6 where PIB helps restrict IIW in the lowest level. Please refer to Appendix D for experimental setups.

## 5.6 SUMMARY OF EXPERIMENTS

We made the following observations from our experiments:

1. We can clearly identify the fitting-compression phase transition during training through our new information measure, i.e., information stored in weights (IIW). Unlike the representation-based information measure $I(X; Z)$, IIW applies to various activation functions including `ReLU`, `sigmoid`, `tanh`, and `linear`.
2. We further identify that the phase transition applies to deeper and wider architecture. More importantly, deeper model is proved to reach faster fitting and compression than shallow models.
3. Unlike $\ell_2$-norm of weights that rise together with wider models, IIW better illustrates the true model complexity and its generalization gap.
4. IIW can explain the performance drop w.r.t. the degree of label noise. Also, IIW can even identify the generalization gap for models learned from random labels.
5. There might exist an optimal batch size for the minimum generalization gap, which is empirically demonstrated by our experiments.
6. Adopting SGD based on the energy function derived from PAC-Bayes IB enables good inference to the optimal posterior of NNs. This works for practical large networks in the literature.

## 6 CONCLUSION

In this paper, we proposed PAC-Bayes information bottleneck and the corresponding algorithm for measuring information stored in weights of NNs and training NNs with information principled regularization. Empirical results show the universality of our information measure on explaining NNs, which sheds light on understanding NNs through information bottleneck. We aim to further investigate its performance and develop this into practical NNs for production in the future.

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

# A    PROOF OF LEMMA A.3

Before the proof of Lemma A.3, we need to introduce a lemma by Martens (2020) as:

**Lemma A.1** (Approximation of Hessian Matrix in NNs (Martens, 2020)). *The Hessian matrix of NNs on a local minimum $\hat{\boldsymbol{\theta}}$ can be decomposed based on Fisher information matrix as*

$$\mathbf{H}_{\hat{\boldsymbol{\theta}}} = \mathbf{F}_{\hat{\boldsymbol{\theta}}} + \frac{1}{n} \sum_{i=1}^{n} \sum_{c=1}^{C} [\nabla_{\hat{\mathbf{y}}} \ell_i(\hat{\boldsymbol{\theta}})]_c \mathbf{H}_{[f]_c}, \tag{A.1}$$

*where $C$ is the total number of classes, $\hat{\mathbf{y}}$ is the output (or prediction) of the network given input $\mathbf{x}$, and $\mathbf{H}_{[f]_c}$ is the Hessian of the $c$-th component of $\hat{\mathbf{y}}$. Specifically, for a well-trained NN, we could have $\nabla_{\hat{\mathbf{y}}} \ell_i(\hat{\boldsymbol{\theta}}) \simeq 0$ thus $\mathbf{H}_{\hat{\boldsymbol{\theta}}} \simeq \mathbf{F}_{\hat{\boldsymbol{\theta}}}$.*

We further introduce a lemma of Poisson bootstrapping as

**Lemma A.2** (Poisson Bootstrapping (Efron, 1992; Chamandy et al., 2012)). *Given an infinite number of samples, the bootstrap resampling weight $\xi$ has the property that $\lim_{n \to \infty} Binomial\left(n, \frac{1}{n}\right) = Poisson(1)$. This approximation becomes precise in practice when $n \geq 100$. Also, we know $\mathbb{E}[\xi_i] = 1$ and $Var[\xi_i] = 1$ by the definition of Poisson distribution when $n$ is large enough.*

When bootstrap resampling from dataset $S$, each individual sample $Z_i = (X_i, Y_i)$ has a probability of $\frac{1}{n}$ being picked, causing the weight $\xi_i$ to follow a binomial distribution as $\xi_i \sim Binomial\left(n, \frac{1}{n}\right)$. As a result, all weights $\boldsymbol{\xi}$ follow a multinomial distribution as $\boldsymbol{\xi} \sim Multinomial\left(n, \frac{1}{n}, \frac{1}{n}, \dots, \frac{1}{n}\right)$ with the total number of samples constrained to be $n$. When it comes to big data, i.e., $n$ is prohibitively large, this multinomial resampling thus becomes rather slow. Based on Lemma A.2, we can now start to prove our lemma of approximating oracle prior covariance.

**Lemma A.3** (Approximation of Oracle Prior Covariance). *Given the definition of influence functions (Lemma 1) and Poisson bootstrapping (Lemma A.2), the covariance matrix of the oracle prior can be approximated by*

$$\Sigma_0 = \mathbb{E}_{p(S)} \left[ (\boldsymbol{\theta}_S - \boldsymbol{\theta}_0)(\boldsymbol{\theta}_S - \boldsymbol{\theta}_0)^\top \right] \simeq \frac{1}{K} \sum_{k=1}^{K} \left( \hat{\boldsymbol{\theta}}_{\boldsymbol{\xi}^k} - \hat{\boldsymbol{\theta}} \right) \left( \hat{\boldsymbol{\theta}}_{\boldsymbol{\xi}^k} - \hat{\boldsymbol{\theta}} \right)^\top \simeq \frac{1}{n} \mathbf{H}_{\hat{\boldsymbol{\theta}}}^{-1} \mathbf{F}_{\hat{\boldsymbol{\theta}}} \mathbf{H}_{\hat{\boldsymbol{\theta}}}^{-1} \simeq \frac{1}{n} \mathbf{F}_{\hat{\boldsymbol{\theta}}}^{-1}, \tag{A.2}$$

*where $\mathbf{F}_{\hat{\boldsymbol{\theta}}}$ is Fisher information matrix (FIM); we omit the subscript $S$ of $\hat{\boldsymbol{\theta}}_S$ and $\hat{\boldsymbol{\theta}}_{S,\boldsymbol{\xi}}$ for notation conciseness, and $\boldsymbol{\xi}^k$ is the bootstrap resampling weight in the $k$-th experiment.*

*Proof.* Recall that in the $k$-th bootstrap resampling process, the loss function is reweighted by $\boldsymbol{\xi}_k = (\xi_{k,1}, \xi_{k,2}, \dots, \xi_{k,n})^\top$. Also, we have an influence matrix $\Psi = (\boldsymbol{\psi}_1, \boldsymbol{\psi}_2, \dots, \boldsymbol{\psi}_n)^\top \in \mathbb{R}^{n \times D}$. The original risk minimizer on the full dataset $S$ is

$$\hat{\boldsymbol{\theta}}_S \triangleq \underset{\boldsymbol{\theta}}{\operatorname{argmin}} \frac{1}{n} \sum_{i=1}^{n} \ell_i(\boldsymbol{\theta}), \tag{A.3}$$

and the reweighted empirical risk minimizer (after bootstrapping) is defined by

$$\hat{\boldsymbol{\theta}}_{S,\boldsymbol{\xi}} \triangleq \underset{\boldsymbol{\theta}}{\operatorname{argmin}} \frac{1}{n} \sum_{i=1}^{n} \xi_i \ell_i(\boldsymbol{\theta}), \tag{A.4}$$

where we omit the subscript $k$ for the sake of conciseness. Given the definition of influence function from Lemma 1, the difference between the two risk minizers above, $\hat{\boldsymbol{\theta}}_S$ and $\hat{\boldsymbol{\theta}}_{S,\boldsymbol{\xi}}$, can be written as

$$\hat{\boldsymbol{\theta}}_{S,\boldsymbol{\xi}} - \hat{\boldsymbol{\theta}}_S \simeq \frac{1}{n} \sum_{i=1}^{n} (\xi_i - 1) \boldsymbol{\psi}_i = \frac{1}{n} \Psi^\top (\boldsymbol{\xi} - \mathbf{1}). \tag{A.5}$$

As a result, the oracle prior can be transformed as

$$\Sigma_0 \simeq \mathbb{E}_{p(S)} \left[ (\hat{\boldsymbol{\theta}}_{S,\boldsymbol{\xi}} - \hat{\boldsymbol{\theta}}_S)(\hat{\boldsymbol{\theta}}_{S,\boldsymbol{\xi}} - \hat{\boldsymbol{\theta}}_S)^\top \right] \tag{A.6}$$

$$\simeq \mathbb{E}_{p(S)} \left[ \left( \frac{1}{n} \Psi^\top (\boldsymbol{\xi} - \mathbf{1}) \right) \left( \frac{1}{n} \Psi^\top (\boldsymbol{\xi} - \mathbf{1}) \right)^\top \right] \tag{A.7}$$

$$= \frac{1}{n^2} \mathbb{E}_{p(S)} \left[ \Psi^\top (\boldsymbol{\xi} - \mathbf{1})(\boldsymbol{\xi} - \mathbf{1})^\top \Psi \right]. \tag{A.8}$$

Furthermore, based on the definition of influence function, we know $\Psi^\top \mathbf{1} = \mathbf{1}^\top \Psi = 0$. The term in Eq. (A.8) can be further writen to

$$\Sigma_0 \simeq \frac{1}{n^2}\mathbb{E}_{p(S)}\left[\Psi^\top(\boldsymbol{\xi} - \mathbf{1})(\boldsymbol{\xi} - \mathbf{1})^\top\Psi\right] = \frac{1}{n^2}\Psi^\top\mathbb{E}_{p(S)}[\boldsymbol{\xi}\boldsymbol{\xi}^\top]\Psi. \tag{A.9}$$

From Lemma A.2 we know $\mathbb{E}[\xi_i] = 1$ and $\mathrm{Var}[\xi_i] = 1$ when $n \geq 100$. We also know that

$$\mathbb{E}_{p(S)}[\xi_i\xi_j] = \begin{cases} \mathbb{E}_{p(S)}[\xi_i]\mathbb{E}_{p(S)}[\xi_j] = 1, & i \neq j, \\ \mathbb{E}_{p(S)}[\xi_i^2] = \mathrm{Var}[\xi_i] + \mathbb{E}^2[\xi_i] = 2, & i = j. \end{cases}$$

This gives rise to the final solution that

$$\Psi^\top\mathbb{E}_{p(S)}[\boldsymbol{\xi}\boldsymbol{\xi}^\top]\Psi = \Psi^\top\left(\mathbf{1}\mathbf{1}^\top + \mathbf{I}_n\right)\Psi \tag{A.10}$$

$$= \sum_{i=1}^n \boldsymbol{\psi}_i\boldsymbol{\psi}_i^\top \tag{A.11}$$

$$= \sum_{i=1}^n \mathbf{H}_{\hat{\boldsymbol{\theta}}}^{-1}\nabla_{\boldsymbol{\theta}}\ell_i(\hat{\boldsymbol{\theta}})\ell_i^\top(\hat{\boldsymbol{\theta}})\mathbf{H}_{\hat{\boldsymbol{\theta}}}^{-1} \tag{A.12}$$

$$= n\mathbf{H}_{\hat{\boldsymbol{\theta}}}^{-1}\left[\frac{1}{n}\sum_{i=1}^n \nabla_{\boldsymbol{\theta}}\ell_i(\hat{\boldsymbol{\theta}})\nabla_{\boldsymbol{\theta}}\ell_i^\top(\hat{\boldsymbol{\theta}})\right]\mathbf{H}_{\hat{\boldsymbol{\theta}}}^{-1} \tag{A.13}$$

$$= n\mathbf{H}_{\hat{\boldsymbol{\theta}}}^{-1}\mathbf{F}_{\hat{\boldsymbol{\theta}}}\mathbf{H}_{\hat{\boldsymbol{\theta}}}^{-1} \tag{A.14}$$

$$\simeq n\mathbf{F}_{\hat{\boldsymbol{\theta}}}^{-1}. \tag{A.15}$$

$\mathbf{I}_n$ in Eq. (A.10) is an identity matrix with size $n \times n$; Eq. (A.11) is true by re-applying the property of influence functions that $\Psi^\top\mathbf{1} = \mathbf{1}^\top\Psi = 0$; and Eq. (A.15) is achieved by the result from Lemma A.1. Summarizing all the above results, the oracle prior covariance can be approximated by

$$\Sigma_0 \simeq \frac{1}{n^2}\left(n\mathbf{F}_{\hat{\boldsymbol{\theta}}}^{-1}\right) = \frac{1}{n}\mathbf{F}_{\hat{\boldsymbol{\theta}}}^{-1}. \tag{A.16}$$

$\square$

# B  PROOF OF LEMMA 3

**Lemma B.1** (Optimal Posterior for PAC-Bayes Information Bottleneck)**.** *Given an observed dataset $S^*$, the optimal posterior $p(\mathbf{w}|S^*)$ of PAC-Bayes IB in Eq. (5) should satisfy the following form that*

$$p(\mathbf{w}|S^*) = \frac{1}{Z(S)}p(\mathbf{w})\exp\left\{-\frac{1}{\beta}\hat{L}_{S^*}(\mathbf{w})\right\} = \frac{1}{Z(S)}\exp\left\{-\frac{1}{\beta}\left(U_{S^*}(\mathbf{w})\right)\right\}, \tag{B.1}$$

*where $U_{S^*}(\mathbf{w})$ is the energy function defined by*

$$U_{S^*}(\mathbf{w}) = \hat{L}_{S^*}(\mathbf{w}) - \beta\log p(\mathbf{w}), \tag{B.2}$$

*and $Z(S)$ is the normalizing constant.*

*Proof.* Recap that the PAC-Bayes information bottleneck in Eq. (5) is

$$\min_{p(\mathbf{w}|S)} \mathcal{L}_{\mathrm{PIB}} = L_S(\mathbf{w}) + \beta I(\mathbf{w}; S). \tag{B.3}$$

Given an observed dataset $S^*$, our object of interest is to find the optimal posterior $p(\mathbf{w}|S^*)$ that minimizes the $\mathcal{L}_{\mathrm{PIB}}$. Consider a constraint of posterior distribution that

$$\int p(\mathbf{w}|S)d\mathbf{w} = 1, \ \forall S \sim p(X,Y)^{\otimes n}, \tag{B.4}$$

we can formulate the problem by

$$\min_{p(\mathbf{w}|S)} \mathcal{L}_{\mathrm{PIB}} = L_S(\mathbf{w}) + \beta I(\mathbf{w}; S),$$

$$\text{s.t.} \int p(\mathbf{w}|S)d\mathbf{w} = 1. \tag{B.5}$$

A Lagrangian can hence be built to relax the above optimization problem by

$$
\begin{aligned}
\min_{p(\mathbf{w}|S)} \widetilde{\mathcal{L}}_{\text{PIB}} &= L_S(\mathbf{w}) + \beta I(\mathbf{w}; S) + \int \alpha_S \int \left( p(\mathbf{w}|S) - 1 \right) d\mathbf{w} dS \\
&= \int p(\mathbf{w}|S) \left[ \hat{L}_S(\mathbf{w}) \right] d\mathbf{w} + \beta \int p(\mathbf{w}, S) \left[ \log p(\mathbf{w}|S) - \log p(\mathbf{w}) \right] d\mathbf{w} dS \\
&\quad + \int \alpha_S \int \left( p(\mathbf{w}|S) - 1 \right) d\mathbf{w} dS,
\end{aligned}
\tag{B.6}
$$

with $\alpha = \{\alpha_S | \forall S \sim p(X, Y)^{\otimes n}\}$ corresponding to Lagrange multipliers; we denote the empirical risk by $\hat{L}_S(\mathbf{w}) = \frac{1}{n} \sum_{i=1}^{n} \ell_i(\mathbf{w})$.

Differentiating $\widetilde{\mathcal{L}}_{\text{PIB}}$ w.r.t. $p(\mathbf{w}|S^*)$ results in

$$
\nabla_{p(\mathbf{w}|S^*)} \widetilde{\mathcal{L}}_{\text{PIB}} = \hat{L}_{S^*}(\mathbf{w}) + \beta \log p(\mathbf{w}|S^*) - \beta \log p(\mathbf{w}) + \beta + \alpha_{S^*}.
\tag{B.7}
$$

Setting $\nabla_{p(\mathbf{w}|S^*)} \widetilde{\mathcal{L}}_{\text{PIB}} = 0$ and solving for $p(\mathbf{w}|S^*)$ yields

$$
\begin{aligned}
\log p(\mathbf{w}|S^*) &= -\frac{1}{\beta} \hat{L}_{S^*}(\mathbf{w}) + \log p(\mathbf{w}) - 1 - \frac{\alpha_{S^*}}{\beta} \\
p(\mathbf{w}|S^*) &= p(\mathbf{w}) \exp \left\{ -\frac{1}{\beta} \hat{L}_{S^*}(\mathbf{w}) \right\} \exp \left\{ -1 - \frac{\alpha_{S^*}}{\beta} \right\}.
\end{aligned}
\tag{B.8}
$$

The second exponential term $\exp \left\{ -1 - \frac{\alpha_{S^*}}{\beta} \right\}$ is the partition function that normalizes the posterior distribution. Denoting the normalization term as $Z(S)$, we hence obtain the optimal posterior solution as

$$
\begin{aligned}
p(\mathbf{w}|S^*) &= \frac{1}{Z(S)} p(\mathbf{w}) \exp \left\{ -\frac{1}{\beta} \hat{L}_{S^*}(\mathbf{w}) \right\} \\
&= \frac{1}{Z(S)} \exp \left\{ -\frac{1}{\beta} \left[ \hat{L}_{S^*}(\mathbf{w}) - \beta \log p(\mathbf{w}) \right] \right\}.
\end{aligned}
\tag{B.9}
$$

$\square$

## C  COMPUTATION FOR PIB OBJECT

Our Alg. 2 presents the training process based on the proposed PIB objective function. It differs from vanilla SGD on two aspects: (1) the objective function (also called energy function $U$) consists of the negative log-likelihood plus a regularization term $\log p(\mathbf{w})$ (line 2); (2) the parameter $\mathbf{w}$ gets update by the gradient of the energy function plus an isotropic Gaussian noise (line 3). Since (2) has no additional computational cost, the major change is the regularization term $\log p(\mathbf{w})$ in (1), described in Eq. (18) as

$$
-\log p(\mathbf{w}) \propto (\mathbf{w} - \boldsymbol{\theta}_0)^\top \Sigma_0^{-1} (\mathbf{w} - \boldsymbol{\theta}_0) + \log(\det \Sigma_0).
\tag{C.1}
$$

The first term is just matrix-vector product based on the approximation of $\Sigma_0$ in Eq. (13). And the second term can be written by

$$
\log(\det \Sigma_0) = \sum_{i=1}^{D} \log \lambda_i,
\tag{C.2}
$$

where $\lambda_i$ is the eigenvalue of $\Sigma_0$, which can be obtained by efficient eigen-decomposition techniques.

## D  EXPERIMENTAL PROTOCOL

All experiments are conducted on MNIST (LeCun et al., 1998) or CIFAR-10 (Krizhevsky et al., 2009). We design two multi-layer perceptron (MLPs): MLP-Small and MLP-Large, where MLP-Small is a two-layer NN, i.e., 784(3072)-512-10, and MLP-Large is a five-layer NN, i.e., 784(3072)-100-80-60-40-10. The number of input units is 784 with permutation MNIST inputs or 3072 with

permutation CIFAR-10 inputs. For the performance comparison in §5.5, we use a reduced version of VGG-Net where two blocks are cut due to the memory constraint. In the general setting, we pick Adam optimizer (Kingma & Ba, 2014) to accelerate the convergence of NN training. We use one RTX 3070 GPU for all experiments.

Specifically for the Bayesian inference experiment, the batch size is picked within $\{8, 16, 32, 64, 128, 256, 512\}$; learning rate is in $\{1e^{-4}, 1e^{-3}, 1e^{-2}, 1e^{-1}\}$; weight decay of $\ell_2$-norm is in $\{1e^{-3}, 1e^{-4}, 1e^{-5}, 1e^{-6}\}$; noise scale of SGLD is in $\{1e^{-4}, 1e^{-6}, 1e^{-8}, 1e^{-10}\}$; $\beta$ of PAC-Bayes IB is in $\{1e^{-1}, 1e^{-2}, 1e^{-3}\}$; and the dropout rate is fixed as $0.1$ for the dropout regularization.

