# OpenReview forum: "PAC-Bayes Information Bottleneck"
_ICLR.cc/2022/Conference — ICLR 2022 Spotlight_

### Official Review · Reviewer_gEEc · 2021-11-01

**Correctness:** 3
**Technical Novelty And Significance:** 4
**Empirical Novelty And Significance:** 3
**Recommendation:** 6
**Confidence:** 3

**Main Review:**

The paper is generally well written and treats an interesting and timely topic. The idea to limit the information about the sample that is contained in the weights is not new (the authors cite several works that bound the generalization error via this information), but this is the first time that I have seen a corresponding cost function implemented in practice. There are, however, a few issues that are not perfectly clear to me:

- The authors cite the literature stating that the generalization gap is limited by I(S;W) if the loss is sigma-sub-Gaussian. Does this hold for the negative log-likelihood in (6)? Also, in (6) is S a random variable or not? (4) requires that I(S;W) is computed as an expectation over p(S), while the log-likelihood in (6) is an expectation over P(w|S), i.e., not over p(S) but over a concrete S. How can this be understood?
- Connected to this, is it safe to call the resulting cost function an information bottleneck cost function? I assume that this is better called an IIW-regularization rather than an IB cost. The IB cost is a very specific formulation that combines a mutual information cost with a mutual information utility, whereas here we have a general cost with an additional mutual information cost as regularization term.
- The authors correctly claim that I(X;T) becomes trivial if the network is deterministic. More precisely, this mutual information becomes infinite in many of these cases (see "Learning Representations for Neural Network-Based Classification Using the Information Bottleneck Principle" by Amjad and Geiger). I believe that this result carries over to I(S;W) being infinite for deterministic learning algorithms. This may not hold for all learning algorithms, but certainly for some. My own gut feeling suggests that I(S;W) is infinite for SGD with finitely many epochs (e.g., by the fact that there are only combinatorially many options to shuffle the batches), but that it is finite for SGLD, where noise is added to the weights. It is therefore not clear to me in which settings the analysis in Section 3 is a valid approximation. In other words, in which settings is the assumption that p(w|S) is Gaussian valid? Does it only hold for SGLD?
- Connected to the point above: In which cases is the assumption that p(w) is Gaussian a valid approximation?
- Can this Gaussian assumption about p(w) be used to bound I(S;W) from above? (E.g., for a Gaussian learning algorithm, can it be shown that the term I(S;W) is maximized if W becomes Gaussian as well? This would be at least intuitive from a channel coding perspective, where a Gaussian channel input is known to maximize the mutual information through a Gaussian channel, and which is then known to produce a Gaussian channel output.)
- In Algorithms 1, pls. compare line 9 with your equation (15). In (15), you sum over squared inner products. In line 9 and 11, you square over the resulting sum of inner products. Is this difference intended, and if so, how can it be explained? Also, do we have $T_0 \ge T_1$ in Algorithm 1?
- In Fig. 1, why is the mutual information I(W;S) evaluated for different layers? What is the exact meaning of splitting the IIW between layers in terms of the generalization bound? I was assuming that the generalization bounds all consider the entire set of weights, and that the proposed PIB should do so as well.
- Also in Fig. 1, the discussion of the inflection point is not fully clear.
- In Section 5.1, it is claimed that the variance of the information explodes. Can this be made more precise (e.g., by writing down the mathematical symbol for this variance)? Furthermore, this is not shown in the figures, if I remember correctly.
- In all figures, why is the mutual information I(S;W) so small? These numbers do not seem right. I would assume that it is necessary to "learn" more than 10⁻2 bits/nats to successfully solve a classification problem. In other words, while the general trend of IIW seems to be correct, I am not convinced of the correctness of the absolute numbers. Can you provide some intuition about these small numbers? Is this connected with the proportionality symbol in (14)? (But going from (8) to (9) it seems to be that additive constants are dropped, not multiplicative constants.)

For the sake of clarity, I would prefer that footnote 3 is in the main text. Also, in some instances the notation and terminology is not clear. E.g., is S sampled iid in (4)? Why is the "oracle prior" called an oracle? How exactly is the bootstrapping resampling weight \zeta_k defined? Why is the temperature $\beta$ called the annealing temperature just before (18)? At the end of Section 5.2 you write that the l2-norm keeps increasing -- the norm of what?

**Summary Of The Paper:**

The authors propose a formulation of the information bottleneck problem, replacing the mutual information between input X and latent representation Z via the mutual information between the sample S and the weights W obtained from the sample. They derive closed-form solutions for this mutual information in the Gaussian setting and propose an SGLD scheme to optimize the objective. Using this objective and optimization algorithm, the authors investigate several interesting scenarios, including different activation functions and noisy labels.

**Summary Of The Review:**

A very interesting paper, dealing with an interesting and timely topic. Unfortunately, the paper is not perfectly clear throughout all sections.

---

> ### Author Response · Authors · 2021-11-15
> **To Reviewer gEEc (1/2)**
>
> ###  **1. Does sigma-sub-Gaussian hold for negative log-likelihood?**
>
> It is guaranteed that a bounded loss function $\ell(\cdot)\in[a,b]$ satisfies $(b-a)/2$-sub-Gaussian (Xu \& Raginsky, 2017). In practice, the easiest way to fulfill it is to clip the negative log-likelihood loss to $[0,b]$ such that it satisfies this condition. We add it to footnote 1 of the new version.
>
> ### **2. Is $S$ in Eq. (6) a random variable?**
>
> $Z$ is a random vector composed of $X$ and $Y$ with $n$ realizations $Z_1,\dots,Z_n$ and has the joint distribution of $X$ and $Y$. $S$ is a random variable that consists of finite-sample $Z$.
>
> ### **3. Difference between Eq. (4) and Eq. (6)**
>
> Eq. (4) depicts a generalization bound of the expectation over $p(S)$. However, in almost all practical cases we only get one concrete dataset $S$ and minimize the empirical risk over it. Such that we design our bottleneck based on the empirical risk (Eqs. (5) and (6)).
>
> ### **4. Why Eq. (6) is called information bottleneck?**
>
> The sufficiency of learned parameters measured by mutual information is
> $$
>     I(W;Y|X,S) = \int p(W|X,S)p(Y|W,X,S)\log \frac{p(Y|W,X,S)}{p(Y|X,S)} \\
>     = E_{p(W|S)}[\text{KL}(p(Y|W,X) \parallel p(Y|X))]  \\
>     = E_{p(W|S)}L_S(W).
> $$
> So Eq. (5) and (6) are other forms of the bottleneck between this sufficiency mutual information term $I(W;Y|X,S)$ and the IIW term $I(w;S)$. We will add this explanation to the new version.
>
> ### **5. Does $I(w;S)$ hold for SGD?**
>
> Since $p(w|S)$ describes an algorithm $A: S\mapsto W$, it differs from the $p(T|X)$ which is regarded as the neural network $T=f(X)$. Unless we inject noise into the network or do generative modeling, the mapping $f:X\mapsto T$ is deterministic and $I(X;T)$ becomes infinite in many cases. On the contrary, modeling on $p(w|S)$ avoids this limitation because SGD itself is one of the randomized algorithms. In this sense, we think $I(w;S)$ holds for SGD as well. We believe it is quite interesting to explore the boundary where $I(w;S)$ becomes trivial in future work.
>
> ### **6. Gaussian assumption of $p(w|S)$ and $p(w)$.**
>
> Because of the complicated learning dynamics of NNs, it is common to make Gaussian assumptions either for variational inference (Burgess et al., 2018) or for learning capability analysis (Rivasplata et al., 2018), for the sake of tractability. It is still an open problem and some recent works are working on improving the bound by taking the data distribution into consideration, e.g., $I(w;S)$ considers $p(S)$ so it becomes more tightened.
>
> ### **7. Bound $I(w;S)$.**
>
> Gaussian distribution is maximum entropy distribution so can we safely assume this as an upper bound for whatever distribution. In information theory, the channel capacity of a Gaussian channel $Y=X+Z$ is reached when the input $X\sim \mathcal{N}(0,\sigma^2)$. But in our case, the input is $S$ and the output is $w$. We may prefer to control the mapping $p(w|S)$ instead of the input distribution. We believe it would be interesting to find the worst-case depending on data distribution where $I(W;S)$ reaches the maximum in future work.
>
> ### **8. Alg. 1 and Eq. (15).**
>
> Thanks for pointing out the typos on line 9 and line 11 of Alg. 1. They should be $\Delta \theta^{\top} \nabla L_t \to (\Delta \theta^{\top} \nabla L_t)^2$ and $\Delta F^2_{T_1} \to \Delta F_{T_1}$. In general, we can keep $T_1=T_0$ by just retrieving the stored gradients of all batches. Or we can do sampling from the stored gradients such that $T_1 < T_0$.
>
> ### **9. IIW in different layers.**
>
> $I(w;S)$ is the information contained in the parameters of the model. We can definitely measure it with the entire set of weights, as we did in Figs. 3, 4, and 5. In Fig. 1 and 2, we separately evaluate the IIW layer-wise for a better display of the dynamics of each layer. This is equivalent to factorizing IIW as
> $$
>     I(W;S) = I(W_1,\dots,W_L;S) \\
>     = \sum_{l=1}^L I(W_l;S|W_1,\dots,W_{l-1}) \\
>      = \sum I(W_l;S),
> $$
> where the last equation holds if we assume each layer's parameters are independent. By this assumption, we could evaluate the IIW within each layer.
>
> ### **10. Inflection point in Fig. 1.**
>
> In Fig. 1, we hope to demonstrate that the epoch where IIW reaches the peak is around the epoch where training loss reaches the inflection point. That is, the training loss has a positive acceleration before this point and then the acceleration turns to be negative. It is pretty interesting to see this happen because the phase transition is reflected both in IIW and in the training loss at the same time point.

---

> ### Author Response · Authors · 2021-11-15
> **To Reviewer gEEc (2/2)**
>
> ### **11. Descriptions in Sec 5.1 for Fig. 1.**
>
> We intend to present the empirical observations that recurrence of memorization phenomenon ($I(w;S)$ starts to increase) after the forgetting phase when we keep training the model for more iterations. From the last figure of Fig. 1, we can see the IIW starts to increase with larger fluctuation after 300 iterations. We think the presentation here is inaccurate. We fix that in the new version.
>
> ### **12. The value of IIW.**
>
> Because of the drop of constant during the approximation (Eq. (8) to Eq. (9)), the approximate IIW $\tilde{I}(w;S)$ no longer holds the same unit as $I(w;S)$, which also makes the absolute value of approximate IIW small. Nonetheless, in our experiments, the value of approximate IIW still succeeds to explain broad aspects of NNs. We believe it would be interesting future work to derive an IIW with nats as its unit.
>
> ### **13. About other minor questions.**
>
> In Eq. (4), $S \sim p(S)$ is sampled i.i.d..
>
> Oracle prior is a terminology drawn from the literature [1]. It is called "oracle" because it minimizes the term $E_{p(S)}[\text{KL}(p(w|S) \parallel p(W)]$ when $p(w)=E_{p(S)}[p(w|S)]$. We will provide the reference in the paper.
>
> We refer the readers to Lemma A.2 (in appendix) about the definition of bootstrapping weight $\xi$ where $\xi \sim \text{Multinomial}(n,\frac1n,\dots,\frac1n)$.
>
> An interesting fact is that the $\beta$ in Eq. (18) and the one in Eq. (5) are the same. When we reduce $\beta$, we both reduce the degree of IIW penalty in the bottleneck and encourage the SGLD to converge. We call it "annealing temperature" following the SGLD literature (Welling \&  Teh, 2011).
>
> We eliminate the ambiguity by "$\ell_2$-norm" to "$\ell_2$-norm of weights" in the new version.
>
>
>
> [1] Dziugaite, G.K., Hsu, K., Gharbieh, W., Arpino, G. \&; Roy, D.. (2021).  On the role of data in PAC-Bayes. Proceedings of The 24th International Conference on Artificial Intelligence and Statistics.

---

### Official Review · Reviewer_CyLM · 2021-11-03

**Correctness:** 4
**Technical Novelty And Significance:** 4
**Empirical Novelty And Significance:** 3
**Recommendation:** 8
**Confidence:** 2

**Main Review:**

**Strengths**
To my knowledge this paper has significant technical and empirical novelty.
The authors do a good job of summarizing previous work and differentiating their contributions.
This is not my area of expertise, but the derivation of PIB, the estimator for $I(\mathbf{w}; S)$, the proposed optimal posterior all look novel and correct.
The experiments are well done, thorough, and support the main claims of the paper.

**Weaknesses**
The main weaknesses of this paper are language and clarity.
I would recommend a thorough Grammarly or perhaps external advice.
The graphs should report means and standard error intervals over multiple random seeds.

**Some specifics**
- Some technical terms are used before definition (e.g., phase transition in abstract)
- In the abstract IB cannot explain, maybe IB theory can, as you use in the intro.
- IIW and $I(\mathbf{w}; S)$ are redundant, would recommend just using $I(\mathbf{w}; S)$
- "Third, mutual information becomes trivial in deterministic cases." Please elaborate / cite.
- "(2) we derive a solution to the intractable...," can something intractable have a solution? maybe approximation is better.
- "optimal posterior of PIB," does PIB have a posterior or is the posterior over the weights?
- Figure 2. IIW only shows compression phase, can the loss also be included in these plots?

**Summary Of The Paper:**

The authors propose new interpretation of the Information Bottleneck (IB), dubbed PAC-Bayes Information Bottleneck (PIB).

Where the IB is defined wrt the mutual information between feature representations $T$ and either inputs $X$ or targets $Y$, PIB is defined wrt the empirical risk over the dataset $S = \\{X_i, Y_i\\}_{i=1}^n$ and the mutual information between model parameters and $S$, $I(\mathbf{w}; S)$, or information stored in weights.

The authors show that PIB is a bound on generalization error.

The authors derive a tractable estimator for $I(\mathbf{w}; S)$.

The authors present an approximate inference method for p(\mathbf{w} \mid S)  that utilizes the proposed PIB.

The authors show that PIB reflects the hypothesized two-phase fitting and compression modes of neural networks across different activation functions, network depth, and network width.

They show that $I(\mathbf{w}; S)$ yields a good estimator of the generalization error that is robust to label noise.

They show that their inference method improves generalization across several benchmark datasets.

**Summary Of The Review:**

To my knowledge this paper demonstrates significant technical and empirical novelty.
I believe the main weaknesses can be addressed prior to publication.
Therefore I recommend acceptance.
However, I am not an expert on this topic, so my confidence is only a 2.

---

> ### Author Response · Authors · 2021-11-15
> **To Reviewer CyLM**
>
> ### **Fitting and Compressing phases in Figure 2**
>
> We guess the reviewer indicates the bottom right figure of Figure 2. We could also observe an initial fitting phase (before the 40 iterations) for the 4-layer MLP although this period is short. The loss and accuracy curves show the similar pattern as in Fig. 1, so we do not put them due to the space limit.
> We also fix the other minor presentation issues proposed by the reviewer in the new version.

---

### Official Review · Reviewer_F2VH · 2021-11-04

**Correctness:** 4
**Technical Novelty And Significance:** 4
**Empirical Novelty And Significance:** 4
**Recommendation:** 10
**Confidence:** 3

**Main Review:**

Strengths:
- The paper proposes an exciting general principle of deep learning. As far as I know, the contributions here are novel and will be of high interest to the community.
- The authors build on previous work by showing how their IB objective addresses the shortcomings of previous work in this area.
- It is very well written. This is a highly-technical paper, and the details are presented in a careful and thoughtful way.
- The experiments are well done and the results support the conclusions. Specifically, this objective is motivated by a PAC-bound (the tightness of which is not clear) and various approximations are used to estimate I(w;S) (the accuracy of these are not immediately clear). The experiments address these issues by showing that the motivation and the approximations are reasonable.

Weaknesses:
- The limitations of this method are not discussed clearly. For example, the paper provides an algorithm for sampling from the weight posterior p(w|S), but how does this compare computationally to standard training of a neural network, or a estimating the posterior in a Bayesian Neural Network?
- There are some minor grammatical and spelling typos throughout, e.g. "infection point".


**Summary Of The Paper:**

This paper proposes a new version of the Information Bottleneck objective for training neural networks. This is in part motivated by previously-derived PAC-Bayes bounds on the generalization error that are proportional to the square root of the mutual information of the weights and the training dataset: I(w;S). Thus this new information bottleneck objective attempts to minimize both the empirical risk and this mutual information.The paper derives a computationally-tractable algorithm for estimating I(w;S), then this algorithm is used to show that this quantity is inversely correlated with generalization loss on a variety of neural network architectures.

**Summary Of The Review:**

An excellent paper with exciting ideas, clear presentation, and technical depth.

---

> ### Author Response · Authors · 2021-11-15
> **To Reviewer F2VH**
>
> ### **Computational comparison of our method to standard SGD NNs and Bayesian NNs.**
>
> Our Alg. 2 presents the training process based on the proposed PIB objective function. It differs from vanilla SGD on two aspects: (1) the objective function (also called energy function $U$) consists of the negative log-likelihood plus a regularization term $\log p(w)$ (line 2); (2) the parameter $w$ gets updated by the gradient of the energy function plus an isotropic Gaussian noise (line 3). Since (2) has no additional computational cost, the major part of change is the regularization term $\log p(w)$ in (1), described in Eq. (18) as
> $$
>         - \log p(w) \propto (w - \theta_0)^{\top} \Sigma_0^{-1} (w - \theta_0) + \log(\det \Sigma_0).
> $$
> The first term is just matrix-vector product based on the approximation of $\Sigma_0$ in Eq. (13). And the second term can be written by
> $$
>     \log (\det \Sigma_0) = \sum_{i=1}^D \log \lambda_i,
> $$
> where $\lambda_i$ is the eigenvalue of $\Sigma_0$, which can be obtained by efficient eigen-decomposition techniques. We will add the discussion regarding to the computational cost in the new version.
>
> Our method and the most of BNNs assume the weights of NNs following Gaussian distribution. BNNs often learn the parameters of the Gaussian (mean, covariance) during the course of training by variational inference. For the sake of computational cost, they have to simplify the covariance as isotropic ($\Sigma=\sigma * I_D$), or as diagonal matrix ($\Sigma=\text{diag}(\sigma_1,\dots,\sigma_D)$). Even with this simplification, it is still burdensome to run Monte Carlo sampling during variational inference to make an estimate of the objective functions (evidence lower bound, ELBO). On the contrary, our method gives a closed-form of the Gaussian parameters, which can be calculated efficiently.

---

### Official Review · Reviewer_HBHz · 2021-11-04

**Correctness:** 3
**Technical Novelty And Significance:** 2
**Empirical Novelty And Significance:** 2
**Recommendation:** 6
**Confidence:** 3

**Main Review:**

Strength:

This paper addresses an interesting and important problem. The proposed PIB is novel. The experiments show that the proposed \tilde{I}(w;S) correlates with the generalization gap, and helps improving the performance.

Weakness:
In order to make the computation of I(w;S) tractable, the authors make several important assumptions. It would strengthen the paper a lot if the paper discuss and perform experiments to show if the assumptions are valid, in the experiments the authors run.

Furthermore, in section 5.5, can the authors show the generalization gap (together with the train and test acc), with different regularization? Ideally we should see that with PIB as the objective, the generalization gap is much smaller than the other methods. With this, we can then be confident that the improvement is due to reduced generalization gap instead of better training.

**Summary Of The Paper:**

This paper introduces the PAC-Bayes Information Bottleneck (PIB). Starting from the generalization bound Eq. 4 which shows that the generalization gap is upper bounded by a function of I(w;S), the authors proposes PIB which has an additional regularization term of \beta I(w;S). Since the computation of I(w;S) is intractable, the authors then make several assumptions to simplify its computation, arriving at an estimate of I(w;S) by \tilde{I}(w;S) (eq. 15), and use SGLD to compute it in practice. Experiments show that (1) there is a two-phase transition in SGD training as indicated by \tilde{I}(w;S); (2) \tilde{I}(w;S) seems to correlate with the generalization gap, under different variations of experiment hyperparameters: number of hidden layers, noise ratio, and number of random label data; (3) it improves performance compared to l2 and dropout regularization.

**Summary Of The Review:**

In summary, this paper is novel, but the experiment should be strengthened as detailed in the main review.

---

> ### Author Response · Authors · 2021-11-15
> **To Reviewer HBHz**
>
> ### **Assumptions made in the paper.**
>
> There are two main assumptions in the paper: (1) the prior and posterior being Gaussian such that the KL-divergence between them has a closed-form solution; (2) influence function is used to approximate parameter difference after bootstrapping the training data.
>
> The assumption (1) is frequently made in the literature because of the complicated learning dynamics of NNs, e.g., in variational inference (Burgess et al., 2018) and in learning capability analysis (Rivasplata et al., 2018, Dziugaite et al., 2019), for the sake of tractability. In this paper, we make this assumption for the same reason. In the future, we plan to address this problem by testing heavy-tailed distributions such as generalized Gaussian distributions and multi-modal distributions such as Gaussian mixtures.
>
>  Influence function is a well-developed tool from robust statistics (Cook \& Weisberg, 1982) and has been proved effective theoretically and empirically in (Koh \& Liang, 2017). In this work, we utilize it to estimate the parameters of the oracle prior $p(w)$. We would like to refer the readers to these literature for the discussion of influence functions.
>
>  Overall, we make the above assumptions for computational benefits. And our experiments show that they get reasonable results.
>
> ### **Track IIW in experiments**
>
> This paper argues that IIW $I(w;S)$ reveals the source of NNs' generalization. And PIB-based training ought to obtain better performance by reducing the generalization gap, which should be demonstrated by smaller IIW in experiments.
>
> Therefore, we add experiments on CIFAR-10 and keep tracking the IIW for four cases: vanilla, $\ell_2$-norm, dropout, and PIB. The obtained IIWs are 5.5e-4, 1.8e-4, 1.9e-4, and 6.4e-5. It is seen that PIB reaches much lower IIW than others, which verifies our claim. We attach this result in the new Fig. 6 in the new version.

---

### Public Comment · ~Omar_Rivasplata1 · 2021-11-10
**sub-gaussian requirement vs plain cross-entropy loss**

Interesting work!

The cost function in Eq. (5) is derived from the bound of Xu & Raginsky in Eq. (4), which requires sub-gaussian losses, is that correct? While the experiments on some benchmark classification datasets are with the plain cross-entropy, is that right? One quick question: How do the authors argue that the plain cross-entropy is sub-gaussian?

---

> ### Author Response · Authors · 2021-11-10
> **Response**
>
> Thanks for your good question! It is guaranteed that a bounded loss function $\ell(\cdot)\in[a,b]$ satisfies $(b-a)/2$-sub-Gaussian (Xu \& Raginsky, 2017). In practice, the easiest way to fulfill it is to clip the negative log-likelihood loss to $[0,b]$ such that it satisfies this condition. We add it to footnote 1 of the new version.

---

> > ### Public Comment · ~Omar_Rivasplata1 · 2021-11-10
> > **clipped cross-entropy vs plain cross-entropy**
> >
> > I see, thanks for your response!
> >
> > Then, this "clipped cross-entropy" is done the same way as in the works of Dziugaite & Roy [1, 2] and in my own work with Perez-Ortiz et al. [3]. It might be misleading when you write "plain cross-entropy loss" considering that the clipping is a significant change?
> >
> > [1] Computing nonvacuous generalization bounds for deep (stochastic) neural networks with many more parameters than training data.
> >
> > [2] Data-dependent PAC-Bayes priors via differential privacy.
> >
> > [3] Tighter risk certificates for neural networks.

---

> > > ### Author Response · Authors · 2021-11-10
> > > **Response**
> > >
> > > During the training, we can monitor to keep the loss below a certain large threshold. However, if the loss keeps decreasing during the course of learning, we actually do not need to do anything. In our experiments, since the loss is stable and never explodes, we did not do clipping for it. Considering it, we do not think this sub-Gaussian assumption will cause significant change to the training in general cases. We will clarify it in the paper. Thanks for your advice!

---

### Public Comment · ~Omar_Rivasplata1 · 2021-11-15
**Did not find the PAC-Bayes guarantee**

After a closer look at this work, I am puzzled by the use of the name "PAC-Bayes" as I did not find the promised PAC-Bayes guarantee.

In fact, I did not find a guarantee at all. From what I see, objective (5) is constructed heuristically and then later the experiments explore the properties of neural network models trained by optimizing this objective. Then the title and claims of guarantees are misleading.

Note that PAC-Bayes bounds are special kinds of PAC (Probably Approximately Correct) bounds, i.e. they are high-probability bounds. The bound of Xu & Raginsky (2017) [their Theorem 1 more precisely] is for the expected gap, where the expectation is over the draw of random dataset $S \sim \mu^{\otimes n}$ and random weight $W \sim P_{W|S}$. This is not a PAC-Bayes bound (they don't call it so, hence the way it is attributed to them as "PAC-Bayes" is misleading). As a matter of fact, the said bound of Xu & Raginsky isn't used in constructing the objective (5). There is a logical gap here: Objective (5) is not derived from Xu & Raginsky's bound in (4). Then, it is unclear why to invoke this result? The only connection I see is that both your objective (5) and Xu & Raginsky's quoted bound involve the quantity $I(W;S)$.

---

> ### Author Response · Authors · 2021-11-16
> **Response**
>
> Hi Omar, thanks for your good question!
>
> Although Xu & Raginsky (2017) do not call their bound PAC-Bayes bound, this bound is indeed a variant of the data-dependent PAC-Bayes bound. Take a linear PAC-Bayes bound based on KL-divergence, like [1]
> $$
> L(w) \leq \frac{1}{1-\frac1{2\lambda}}(L_S(w) + \frac{\lambda L_{max}}{m}(\text{KL}(p(w|S)\|p(w))+\log\frac1\delta),
> $$
> for example. We can pick an "oracle prior" to **minimize the risk bound in expectation** proved by Theorem 2.2 in [2] (drawn from [3]). In detail, the oracle prior is
> $$
> p(w) \triangleq E_{S\sim p(S)}[p(w|S)].
> $$
> Now we can see that the KL-divergence term in expectation becomes
> $$
> E_{p(S)}[\text{KL}(p(w|S)\|p(w))] = I(w;S)
> $$
> which is the mutual information term called information in weights (IIW) throughout this paper. This data-dependent prior hence gives us the tightest linear PAC-Bayes bound in expectation.
>
> In this paper, we refer to (Xu & Raginsky, 2017) because they are the one of earliest works approaching deep learning analysis based on IIW term. This work gives rise to a series of works on using IIW to bound NNs, which are all based on this PAC-Bayes literature [4,5].
>
> You are welcome to refer to point 3 & 4 in the response to Reviewer gEEc why we take the empirical risk on realized $S$ instead of the expectation in the PIB objective.
>
>
>
> [1] McAllester, D. A. A PAC-Bayesian Tutorial with a Dropout Bound, 2013.
>
> [2] Dziugaite, G. K., Hsu, K., Gharbieh, W., Arpino, G., & Roy, D. (2021, March). On the role of data in PAC-Bayes. In *International Conference on Artificial Intelligence and Statistics* (pp. 604-612). PMLR.
>
> [3] John Langford and Avrim Blum (2003). “Microchoice bounds and self bounding learning
> algorithms”. Machine Learning 51.2, pp. 165–179.
>
> [4]  W. Mou, L. Wang, X. Zhai, and K. Zheng. “Generalization Bounds of SGLD for Non-convex Learning: Two Theoretical Viewpoints”. In: Proceedings of the 31st Conference On Learning Theory. Ed. by S. Bubeck, V. Perchet, and P. Rigollet. Vol. 75. Proceedings of Machine Learning Research. PMLR, June 2018, pp. 605–638.
>
> [5] Negrea, J., Haghifam, M., Dziugaite, G. K., Khisti, A., & Roy, D. M. (2019). Information-Theoretic Generalization Bounds for SGLD via Data-Dependent Estimates. *Advances in Neural Information Processing Systems*, *32*, 11015-11025.

---

> > ### Public Comment · ~Omar_Rivasplata1 · 2021-11-18
> > **Response**
> >
> > Hi :)
> >
> > Thank you (!) for engaging the conversation.
> >
> > I think ideally if these things you have described could make their way into the paper, it would help the readers to understand the ideas you're trying to connect in this work, and how. Xu & Raginsky (2017) is a great work and relevant to support the connection to IIW in your work. I continue to think that it is misleading to call their bound a PAC-Bayes bound. There are some works on generalisation bounds based on information-theoretic quantities that might be relevant to support the connection you're trying to make, for instance https://arxiv.org/pdf/2102.02016.pdf  which includes a high-probability bound in terms of $I(W;S)$.
> >
> > Probably you could comment that obtaining a rigorous guarantee for the output of optimising (5) is the topic of future research (If it's believable that it's possible to obtain such guarantee).
> >
> > One correction: The complete title of [2] is "On the role of data in PAC-Bayes bounds."
> >
> > Good luck with your paper!

---

### Author Response · Authors · 2021-11-15
**General Response for All Reviewers**

We thank the reviewer for the time spent in reviewing this paper and the valuable questions which help improve this study a lot. We would like to highlight the motivation and the contribution of this paper here:

Understanding the source of the superior generalization ability of NNs remains one of the most important problems in ML research. There have been a series of theoretical works trying to derive non-vacuous bounds for NNs. Recently, the compression of information stored in weights (IIW) is proved to play a key role in NNs generalization based on the PAC-Bayes theorem. However, no solution of IIW has ever been provided, which builds a barrier for further investigation of the IIW's property and its potential in practical deep learning. In this paper, we propose an algorithm for the efficient approximation of IIW. Then, we build an IIW-based information bottleneck on the trade-off between accuracy and information complexity of NNs, namely PIB. From PIB, we can empirically identify the fitting to compressing phase transition during NNs' training and the concrete connection between the IIW compression and the generalization. Besides, we verify that IIW is able to explain NNs in broad cases, e.g., varying batch sizes, over-parameterization, and noisy labels. Moreover, we propose an MCMC-based algorithm to sample from the optimal weight posterior characterized by PIB, which fulfills the potential of IIW in enhancing NNs in practice.

For the point-to-point answers, please refer to the specific responses below.

---

### Decision · Program_Chairs · 2022-01-20

**Decision:**

Accept (Spotlight)

**Comment:**

This paper revisits the information bottleneck principle, but in terms of the compression inherent in the weights of a neural network, rather than the representation. This gives the resulting IB principle a PAC-Bayes flavor. The key contribution is a generalization bound based on optimizing the objective dictated by this principle, which is then tractably approximated and experimentally verified. Reviews raise concerns about assumptions made to achieve the tractable version and a public discussion debates whether this is truly a PAC-Bayes bound. The authors address these adequately. Another concern is whether improvements in experiments can be ascribed to the new objective. Authors add new experiments in support of this. Additional concerns about the clarity of certain aspects of the paper were or were promised to be addressed by the authors. Overall, the perspective of this paper, its technical contributions, and experimental evaluations appear to be worthwhile to share with the community, as they advance the applicability of the information bottleneck principle.